# VBH-GNN: Variational Bayesian Heterogeneous Graph Neural Networks for Cross-subject Emotion Recognition

**Chenyu Liu**[1†], **Xinliang Zhou**[1†], **Zhengri Zhu**[2], **Liming Zhai**[3*], **Ziyu Jia**[4*], **Yang Liu**[1]

[1]School of Computer Science and Engineering, Nanyang Technological University
[2]School of Electronic and Information Engineering, Beijing Jiaotong University
[3]School of Computer Science, Central China Normal University
[4]Institute of Automation Chinese Academy of Sciences
{chenyu003,xinliang001}@e.ntu.edu.sg, zhengrizhu@bjtu.edu.cn,
limingzhai@ccnu.edu.cn, jia.ziyu@outlook.com, yangliu@ntu.edu.sg

## Abstract

The research on human emotion under electroencephalogram (EEG) is an emerging field in which cross-subject emotion recognition (ER) is a promising but challenging task. Many approaches attempt to find emotionally relevant domain-invariant features using domain adaptation (DA) to improve the accuracy of cross-subject ER. However, two problems still exist with these methods. First, only single-modal data (EEG) is utilized, ignoring the complementarity between multimodal physiological signals. Second, these methods aim to completely match the signal features between different domains, which is difficult due to the extreme individual differences of EEG. To solve these problems, we introduce the complementarity of multi-modal physiological signals and propose a new method for cross-subject ER that does not align the distribution of signal features but rather the distribution of spatio-temporal relationships between features. We design a Variational Bayesian Heterogeneous Graph Neural Network (VBH-GNN) with Relationship Distribution Adaptation (RDA). The RDA first aligns the domains by expressing the model space as a posterior distribution of a heterogeneous graph for a given source domain. Then, the RDA transforms the heterogeneous graph into an emotion-specific graph to further align the domains for the downstream ER task. Extensive experiments on two public datasets, DEAP and Dreamer, show that our VBH-GNN outperforms state-of-the-art methods in cross-subject scenarios.

## 1 Introduction

Emotion is a complex physical and psychological state that plays a vital role in human decision-making, behavior, and interpersonal interaction (Cabanac (2002)). In recent years, emotion recognition (ER) has become increasingly important in fields such as diagnosis of depression and human-computer interaction (Shneiderman et al. (2016); Zhou et al. (2023)). To study emotions, researchers usually record changes in body language (Coulson (2004)), voice (Zeng et al. (2007)), expression (Ekman (1984)), and physiological signals (Gunes et al. (2011)) after inducing emotions in subjects. Since external cues are easy to control, they are not guaranteed to reflect the actual emotional state of the subject. In contrast, physiological signals cannot be disguised, capturing subjects' potential reactions that reveal real human emotions (Mühl et al. (2014)). Electroencephalogram (EEG), as a high-resolution and effective physiological signal, is therefore widely used in emotion recognition (ER) (Calvo & D'Mello (2010); Liu et al. (2024)).

Compared to conventional ER, cross-subject ER is undoubtedly more appealing but also presents greater challenges. First, EEG has high individual differences. Since emotion is a comprehensive reflection of human physiology and psychology, the differences in anatomical structures, biological rhythms, personalities, and psychological states among subjects can lead to different responses to

---

†Co-first Authors, *Corresponding Authors

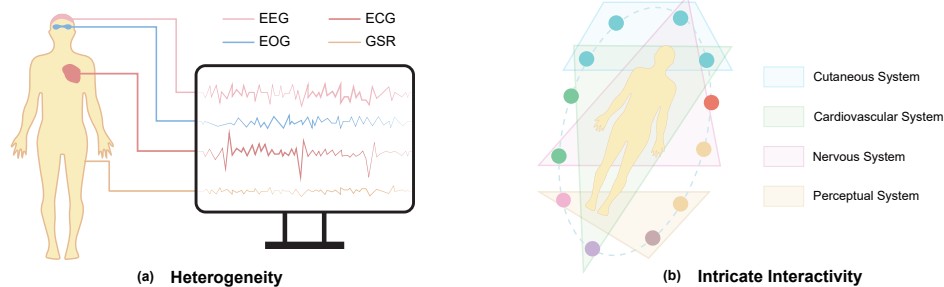

Figure 1: Heterogeneity and intricate interactivity. The heterogeneity means the difference among the signals from different modalities. The intricate interactivity refers to the multi-modal signals belonging to multiple physiological systems that dynamically cooperate to regulate human emotions.

the same emotional stimuli (Wan et al. (2021)). Second, EEG typically has a low signal-to-noise ratio due to internal and external factors. Internally, EEG often contains substantial biological noise from muscle movements. Externally, non-invasive EEG acquisition techniques limit the sampling rate and are susceptible to interference. These reasons restrict the robustness and generalizability of cross-subject ER in practical applications.

Existing work has demonstrated that multi-modal physiological signals can improve the accuracy of ER. Additionally, domain adaptation (DA) approaches have been explored for EEG-based cross-subject ER. However, no studies have yet combined multi-modalities and DA for cross-subject ER due to the intricate interactivity of multi-modal signals (see Figure 1 (b)). Current methods have two limitations: 1) they fail to capture the spatio-temporal relationships between modalities while considering their heterogeneity, and 2) they do not utilize such spatio-temporal relationships to align different domains.

For the first problem, although most methods use modality-specific feature extractors to capture the heterogeneity (see Figure 1 (a)) of multi-modal data (Lan et al. (2020); Ma et al. (2019); Mittal et al. (2020); Wu et al. (2024; 2023a); Wang et al. (2022)), they ignore the spatio-temporal relationship between the modalities during the feature fusion stage. Other methods combine multi-modal signals to capture spatio-temporal relationships (Zhang et al. (2020); Wu et al. (2022; 2023b); Liu et al. (2023); Pinto et al. (2019)), but the heterogeneity between modalities is neglected in the feature extraction process. For the second problem, existing DA methods (She et al. (2023); Peng et al. (2022)) are stuck on completely matching the EEG feature distributions between different domains, which is difficult because of individual differences. They ignore utilizing the spatio-temporal relationships of modalities between subjects to build connections among domains.

To solve these problems, we introduce a new approach to align source and target distributions by multi-modal spatial-temporal relationships to achieve cross-subject ER. We propose Variational Bayesian Heterogeneous Graph Neural Networks (VBH-GNN), which integrate modalities' temporal and spatial Relationship Distribution Adaptation (RDA) between domains and ER into one framework. Specifically, the RDA contains Bayesian Graph Inference (BGI) and Emotional Graph Transform (EGT). The BGI models multi-modal relationships as heterogeneous graphs (HetG) and aligns the relationships of domains via the edge distribution of HetG based on the Variational Bayesian theorem. EGT transforms the HetG into emotion-specific graphs (EmoG), further aligning the source and target domains while differentiating the relationships of modalities under different emotions. After the joint constraints of these two steps, the VBH-GNN can infer the domain-invariant multi-modal spatio-temporal relationships between source and target domains and utilize this relationship to weight the signal feature and feed it to the classifier for cross-subject ER. We perform extensive quantitative and qualitative experiments on multiple datasets, and the results demonstrate that our VBH-GNN significantly outperforms previous state-of-the-art methods.

We make the following main contributions: 1) we introduce a novel deep learning architecture named VBH-GNN that combines multi-modal physiological signals and domain adaptation (DA) for more accurate cross-subject ER. 2) we propose Relationship Distribution Adaptation (RDA) to align source and target domains through multi-modal spatio-temporal relationship distributions. 3) we develop Bayesian Graph Inference (BGI) to model and align intricate interactivity of modali-

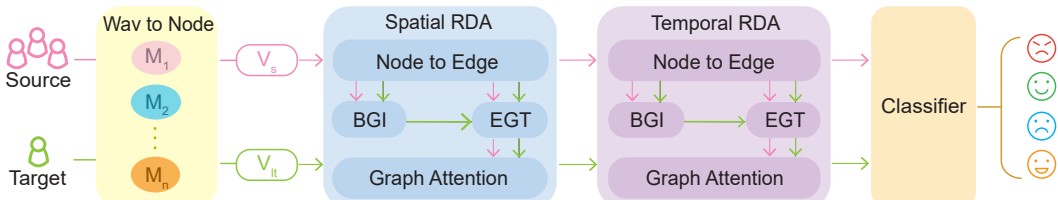

Figure 2: Flowchart of VBH-GNN.

ties in source and target domains. 4) we design a graph learning module named Emotional Graph Transform (EGT) to optimize the aligned domain distribution for downstream ER tasks.

## 2 PRELIMINARIES

**Mulit-modal Domain Adaptation.** This paper focuses on the scenario where the source and target domains have several modal types (*i.e.*, EEG, Electrocardiography (ECG), Galvanic Skin Response (GSR), *etc.*). We denote the source domain as $\mathcal{D}_s = \left\{ \left( X_s^{EEG}, \cdots, X_s^{ECG}, Y_s \right) \right\}$, where $(X_s^{EEG}, \cdots, X_s^{ECG})$ are multi-modal data with label $Y_s$. For the target domain, we donate a limited number of labeled target data by $\mathcal{D}_{lt} = \left\{ \left( X_{lt}^{EEG}, \cdots, X_{lt}^{ECG}, Y_{lt} \right) \right\}$, and unlabelled target data $\mathcal{D}_{ut} = \left\{ \left( X_{ut}^{EEG}, \cdots, X_{ut}^{ECG} \right) \right\}$. The ER model is trained on $\mathcal{D}_s$ and $\mathcal{D}_{lt}$, and evaluated on $\mathcal{D}_{ut}$.

**Heterogeneous Graph.** We define a HetG as G $= (V, E, \phi, \psi)$, where $V$ and $E$ denote the node set and edge set. $\phi : V \to T_V$ maps the node to its corresponding type in node type set $T_V$, and similarly $\psi : E \to T_E$ stands for the edge mapping. For a HetG, $|T_V| + |T_E| > 2$.

## 3 VARIATIONAL BAYESIAN HETEROGENEOUS GRAPH NEURAL NETWORKS

### 3.1 OVERVIEW

As shown in Figure 2, there are four components included in our cross-subject ER process: 1) Wav to Node, 2) Spatial RDA and Temporal RDA, and 3) a Classifier.

**Wav-to-Node** employs the same setup as in (Jia et al. (2021)), transforming multi-modal signals into node embeddings with modal-specific networks to capture the heterogeneity of modalities. It accepts multi-modal signals $\mathcal{D}_s$ and $\mathcal{D}_{lt}$ from the source and target domain as input and converts them into node embeddings $V_s, V_{lt} \in \mathbb{R}^{B \times N_n \times D_n}$:

$$(V_s, V_{lt}) = \bigcup_{i=1}^{M} f_i(X_s^i, X_{lt}^i) \tag{1}$$

where $X_s^i$ and $X_{lt}^i$ represents the signal of modality $i$ in two domains, and $f_i(\cdot)$ represents the modality-specific feature extraction network of modality $i$.

**RDA** is the core component of VBH-GNN. It accepts node embeddings for domain alignment and updates the weights of node embeddings. VBH-GNN contains Temporal RDA and Spatial RDA, which perform inference and alignment of relationship distributions in the temporal and spatial dimensions. The details of RDA will be explained in Section 3.2. The overview process of RDA is as follows:

$$(V_s', V_{lt}') = \text{SRDA}(V_s, V_{lt}) \tag{2}$$

$$(V_s'', V_{lt}'') = \text{TRDA}(f_{\text{TRANS}}(f_{\text{FNN}}((V_s', V_{lt}') + (V_s, V_{lt}))) \tag{3}$$

where SRDA$(\cdot)$ and TRDA$(\cdot)$ represent Spatial RDA and Temporal RDA. $f_{\text{FNN}}$ is a feed-forward network. $f_{\text{TRANS}}(\cdot)$ is an operation that transforms spatial nodes into temporal nodes.

**Classifier** takes the output of the RDA and predicts labels for two domains with a classification network:

$$(Y_s', Y_{lt}') = f_c(V_s'', V_{lt}'') \tag{4}$$

**Loss of VBH-GNN** contains two types of loss: the RDA Loss for aligning the source domain and target domain and the prediction loss of the classifier. The final loss function is formulated as

$$\mathcal{L}_{\text{VBH-GNN}} = \lambda_1 \mathcal{L}_{SRDA} + \lambda_2 \mathcal{L}_{TRDA} + \lambda_3 \mathcal{L}_{SBCE} + \lambda_4 \mathcal{L}_{TBCE} \tag{5}$$

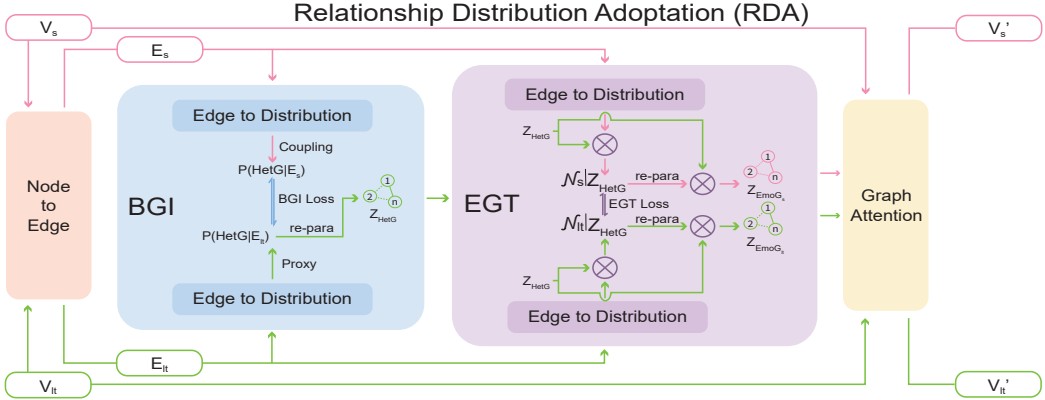

Figure 3: Relationship distribution adaptation (RDA). It contains Node-to-Edge, Bayesian Graph Inference (BGI), Emotional Graph Transform (EGT), and Graph attention (GA).

where $\mathcal{L}_{\text{SRDA}}$ and $\mathcal{L}_{\text{TRDA}}$ are loss of Spatial RDA and Temporal RDA (will be further explained in Section 3.2.3), $\mathcal{L}_{SBCE}$ and $\mathcal{L}_{TBCE}$ are Binary Cross-Entropy Loss for source and target domain classification. $\lambda_1$, $\lambda_2$, $\lambda_3$, and $\lambda_4$ are loss weights, which are all set to 1 in the experiments.

## 3.2 RELATIONSHIP DISTRIBUTION ADAPTATION

RDA implements two of the most critical functions of VBH-GNN, the modeling of multi-modal relationships and the alignment of domains. As shown in Figure 3, it contains four procedures:

**Node-to-Edge** accepts node embeddings as input. It denotes the multi-modal spatio-temporal relationship as a HetG by generating an edge embedding between every two nodes through a convolutional layer $f_{\text{emb}}(\cdot)$:

$$E_s, E_{lt} = f_{\text{emb}}\left(V_s[i,j], V_{lt}[i,j]\right) \quad (i,j \in [1,n]|i! = j) \tag{6}$$

where $V_s, V_{lt} \in \mathbb{R}^{B \times N_v \times D_v}$ represents the input node objects, $V_s[i,j], V_{lt}[i,j] \in \mathbb{R}^{B \times N_e \times 2 \times D_v}$ denotes the combination of two node embeddings, and $E_s, E_{lt} \in \mathbb{R}^{B \times N_e \times D_e}$ represents the edge embedding set.

**BGI** finds the latent relationship distribution of multi-modal physiological signals shared by the source and target subjects. It couples the intricate interactivity between modalities and expresses the HetG edge distribution of the target domain as the posterior distribution of source domain HetG edges based on the Variational Bayesian theorem. The source and target domains are aligned via BGI Loss (see Figure 4(b)). The BGI process is as follows:

$$Z_{\text{HetG}} = \text{BGI}(E_s, E_{lt}) \tag{7}$$

where $Z_{\text{HetG}} \in \mathbb{R}^{B \times N_e}$ is sampling of HetG edge distribution.

**EGT** distinguishs between the latent relationship distribution founded by BGI in different emotions. It transforms the HetG into EmoG that discriminates different emotion distributions in the source and target domains (see Figure 4(c)). The EGT process is as follows:

$$Z_{\text{EmoG}_{s \lor lt}} = \text{EGT}(Z_{\text{HetG}}, E_s, E_{lt}) \tag{8}$$

where $Z_{\text{EmoG}_{s \lor lt}} \in \mathbb{R}^{B \times N_e}$ is the sampling of EmoG edge distribution of source and target domains.

**Graph Attention** (GA) is an attention module that updates the node embedding weights. Specifically, the output of EGT is transformed $\mathcal{T}(\cdot)$ as an adjacency matrix and multiplied with node embedding $V_{s \lor lt}$ that passes through a linear layer $f(\cdot)$. The final output of RDA is defined as

$$V'_{s \lor lt} = \mathcal{T}(Z_{\text{EmoG}_{s \lor lt}}) \times f(V_{s \lor lt}) \tag{9}$$

### 3.2.1 BAYESIAN GRAPH INFERENCE (BGI)

BGI models the alignment of the HetG edge distributions in the source and target domains as a Variational Bayesian inference process. First, we define the intricate interactivity of multi-modal

signals as the coupling of an infinite number of physiological system relationships. Specifically, an infinite number of edges $\{e^n : n \in \mathbb{N}\}$ exist between node $v_i$ and $v_j$, and the likelihood of an edge existing is very small $\forall n \in \mathbb{N}, p^n \to 0$. We therefore define the distribution of $e_{i,j}$ as

$$P(e_{i,j}) = \sum_{n=1}^{\infty} \text{BER}(p^n) \sim \lim_{\substack{n \to \infty \\ p_{i,j} \to 0}} \text{BIN}(n, p_{i,j}) \tag{10}$$

where $\text{BER}(\cdot)$ is the Bernoulli distribution. $\text{BIN}(\cdot)$ is the Binomial Distribution representing the coupling of infinite edges between two nodes. $p_{i,j}$ is the parameters of the Binomial distribution computed by the neural network. From this, we define the prior HetG edge distribution from the source domain as follows:

$$P(\text{HetG}|E_s) \sim \text{BIN}(n, p_s) \tag{11}$$

where $p_s \in \mathbb{R}^{B \times N_e}$ and $n \to \infty$.

Unlike VRNN (Chung et al. (2015)), which applies an RNN encoder to estimate the parameters of the approximate posterior distribution, the posterior distribution of HetG edges cannot be computed similarly due to the presence of infinite $n$. According to De Moivre–Laplace theorem in (Sheynin (1977); Liu & Jia (2022)), the Binomial Distribution $\text{BIN}(n, p_{i,j})$ can be infinitely approximated by a Gaussian distribution $\mathcal{N}(np_{i,j}, np_{i,j}(1 - p_{i,j}))$ with $n \to \infty$. To further circumvent the straight-forward parameterization of $p_{i,j}$ and $n$, we parameterize a Gaussian distribution $\mathcal{N}(\tilde{\mu}_{i,j}, \tilde{\sigma}_{i,j}^2)$, where $\tilde{\mu}_{i,j}$ and $\tilde{\sigma}_{i,j}$ are calculated from $e_{i,j} \in E_{lt}$ by a neural network. First, we define an intermediate variable $\lambda_{i,j}$:

$$\lambda_{i,j} = \frac{1}{1 - 2\tilde{\mu}_{i,j}} \sim \zeta(\tilde{\mu}_{i,j}) + \epsilon \tag{12}$$

where $\lambda_{i,j}$ is an approximation used to prevent explosions in the calculation. $\epsilon$ is a small hyperparameter that is the lower bound for $\lambda_{i,j}$. We define a Gaussian distribution as follows:

$$\mu_{i,j} = \frac{1 + 2\lambda_{i,j}\tilde{\sigma}_{i,j}^2 - \sqrt{1 + 4\lambda_{i,j}^2 \tilde{\sigma}_{i,j}^4}}{2} \tag{13}$$

$$P(e_{i,j}) \sim \mathcal{N}(\mu_{i,j}, \mu_{i,j}(1 - \mu_{i,j})) \tag{14}$$

$P(e_{i,j})$ is a Gaussian proxy to the Binominal distribution $\text{BIN}(n, p_{i,j})$ with a minimal constant divergence. We define the HetG edge distribution of the target domain as the posterior distribution as follows:

$$P(\text{HetG}|E_{lt}) \sim \mathcal{N}(\mu_{lt}, \mu_{lt}(1 - \mu_{lt})) \tag{15}$$

where $\mu_{lt} \in \mathbb{R}^{B \times N_e}$. From this, we can apply re-parametrization (Kingma & Welling (2013)) to draw samples form the distribution of HetG edges:

$$Z_{\text{HetG}} = \sqrt{\mu_{lt}(1 - \mu_{lt})} \times \epsilon + \mu_{lt} \tag{16}$$

### 3.2.2 EMOTIONAL GRAPH TRANSFORM (EGT)

The core concept of EGT is that it transforms innumerable intricate relationships into a recognizable notion of emotion in both source and target domains. EGT learns a Gaussian variable conditioned on the HetG edge to perform a weighted transformation on the HetG edges, and the final graph obtained is the EmoG representing the informative emotion representation in each domain. We assume such Gaussian variable as follows:

$$\mathcal{N}_{s \vee lt}|Z_{\text{HetG}} \sim \mathcal{N}(Z_{\text{HetG}} \times \bar{\mu}_{s \vee lt}, Z_{\text{HetG}} \times \bar{\sigma}_{s \vee lt}^2) \tag{17}$$

where $Z_{\text{HetG}}$ represents the HetG edge sampling from Eq. (16) and is utilized as a conditional restriction that prohibits the EmoG edge from acting arbitrarily when certain HetG edge sample values are near zero. $\bar{\mu}_{s \vee lt}$ and $\bar{\sigma}_{s \vee lt}$ represent the parameters of Gaussian distributions computed from two domains through the neural network during the EGT procedure. Then, we transform HetG by using the sampling of Gaussian variables $\mathcal{N}_{s \vee lt}$ as weights:

$$Z_{\text{EmoG}_{s \vee lt}} = (\sqrt{Z_{\text{HetG}}} \times \bar{\sigma}_{s \vee lt} \times \epsilon + Z_{\text{HetG}} \times \bar{\mu}_{s \vee lt}) \times Z_{\text{HetG}} \tag{18}$$

### 3.2.3 LOSS OF RDA

Loss of RDA measures the difference between both HetG and EmoG edge distribution of the source and target domains. It contains two parts:

$$\mathcal{L}_{\text{RDA}} = \mathcal{L}_{\text{BGI}} + \mathcal{L}_{\text{EGT}} \tag{19}$$

**BGI Loss** is the loss of the BGI procedure. Its practical meaning is the Kullback–Leibler Divergence (KLD) between the prior and posterior distributions of the HetG edges:

$$\mathcal{L}_{\text{BGI}} = KLD\{P(\text{HetG}|E_s)||P(\text{HetG}|E_{lt})\} = KLD\{\text{BIN}(n, p_s)||\mathcal{N}(\mu_{lt}, \mu_{lt}(1 - \mu_{lt}))\} \tag{20}$$

However, since the prior edge distribution of HetG is a Binomial distribution $\text{BIN}(n, p_s)$ which contains an infinite parameter $n$, $\mathcal{L}_{\text{BGI}}$ is computationally intractable. Therefore, we approximate minimizing the loss as minimizing its upper bound. The closed-form solution irrelevant to $n$ is calculated as follows:

$$\text{Min}(\mathcal{L}_{\text{BGI}}) = \text{Min}(\mu_{lt} \log \frac{\mu_{lt} + \epsilon}{p_s + \epsilon} + (1 - \mu_{lt}) \log \frac{1 - \mu_{lt} + \mu_{lt}^2/2 + \epsilon}{1 - p_s + p_s^2/2 + \epsilon}) \tag{21}$$

**EGT Loss** is a further alignment, which forces the informative emotion representation of domains transformed from HetG also remain consistent. It is defined as follows:

$$\mathcal{L}_{\text{EGT}} = KLD\{\mathcal{N}_s|Z_{\text{HetG}}, E_s||\mathcal{N}_{lt}|Z_{\text{HetG}}, E_{lt}\} \tag{22}$$

where $\mathcal{N}_s|Z_{\text{HetG}}$ and $\mathcal{N}_{lt}|Z_{\text{HetG}}$ are the Gaussian variables in Eq. (17). We abbreviate the mean and the standard deviation of these Gaussian variables as $\mu_s$ ($\mu_{lt}$) and $\sigma_s^2$ ($\sigma_{lt}^2$). Therefore, EGT loss could be expressed as

$$\text{Min}(\mathcal{L}_{\text{EGT}}) = \text{Min}(2 \log(\frac{\sigma_s + \epsilon}{\sigma_{lt} + \epsilon}) + \frac{\sigma_{lt}^2 + (\mu_{lt} - \mu_s)^2}{(\sigma_s + \epsilon)^2} - 1) \tag{23}$$

## 4 EXPERIMENTS

### 4.1 EXPERIMENTAL SETTING

**Dataset.** Our experiments were conducted on **DEAP** (Koelstra et al. (2011)) and **DREAMER** (Soleymani et al. (2011)). They are both multi-modal physiological signal datasets tailored for human emotion analysis. Both datasets recorded multiple physiological signals from subjects while watching videos (DEAP contains EEG, Electrooculogram (EOG), Electromyogram (EMG), and GSR, and DREAMER contains EEG and ECG). These videos are designed to elicit different emotional responses characterized by valence (high/low) and arousal (high/low). Subjects will evaluate the valence and arousal with a given scale (DEAP from 1 to 9 and DREAMER from 1 to 5). Detailed dataset information can be found in Appendix A.1. We adopt a general ER paradigm: valence and arousal scores are divided into two categories bounded by median number (For DEAP, it is 5; For DREAMER, it is 3); train and predict valence and arousal, respectively.

**Training strategy.** We adopt the leave-one-subject-out paradigm to divide the source and target domains. The target domain contains one subject, and the remaining subjects are the source domain. This process is repeated for each subject, and the results of all subjects are averaged as the final result. We followed the supervised DA paradigm in (Wang et al. (2022)) to divide the training and testing sets. Specifically, The data of the target subject is divided into 5 folds, where one fold is the training set used as the labeled target domain, another fold is the validation set, and the remaining folds are used as the testing set. The average of all five experiments' results is the target subject's final result. The evaluation metrics we use are accuracy and F1 score (Explained in the Appendix A.2).

**Cropping strategy.** To expand the data volume, our experiments used a cropped strategy (Schirrmeister et al. (2017)), which crops the data of each trail into $4s$ non-overlapping segments. Note that our cropping is done strictly after splitting the training and testing set. Since physiological signals are time-series signals, there is a correlation between neighboring segments. If they appear separately in the training and testing sets, it will cause data leakage. We make all models follow the same cropping strategy to prevent abnormally high accuracy and F1 scores caused by data leakage.

Table 1: Performance comparison of all methods in terms of accuracy and f1 score (in %) for classification tasks (the greater the better) on DEAP and DREAMER. The best, second-best, and third-best results are highlighted in [1], [2], and [3], respectively.

| Baseline | DEAP ( EEG, EOG, EMG and GSR) | | | | DREAMER (EEG and ECG) | | | |
| --- | --- | --- | --- | --- | --- | --- | --- | --- |
| | Arousal | | Valence | | Arousal | | Valence | |
| | Accuracy | F1 Score | Accuracy | F1 Score | Accuracy | F1 Score | Accuracy | F1 Score |
| DGCNN (Song et al. (2018)) | 59.52 ± 9.19 | 61.13 ± 13.67 | 65.97 ± 7.01 | [3]70.8 ± 7.75 | 64.94 ± 9.36 | 63.05 ± 11.92 | 62.19 ± 5.25 | 56.22 ± 8.41 |
| EEGNet (Lawhern et al. (2018)) | 62.28 ± 8.46 | 66.67 ± 10.28 | 63.33 ± 8.03 | 67.62 ± 7.44 | 62.02 ± 10.33 | 57.56 ± 11.69 | 64.71 ± 6.27 | 58.76 ± 9.52 |
| MM-ResLSTM (Ma et al. (2019)) | 68.42 ± 8.66 | 65.95 ± 10.36 | 67.69 ± 7.43 | 61.58 ± 8.08 | 66.23 ± 10.65 | 60.53 ± 11.68 | 67.87 ± 4.47 | [2]70.61 ± 6.52 |
| SST-EmotionNet (Jia et al. (2020)) | 68.54 ± 8.57 | 59.74 ± 9.96 | 66.65 ± 7.22 | 65.7 ± 7.77 | 67.57 ± 8.18 | 61.65 ± 9.92 | 68.58 ± 8.68 | 63.0 ± 6.62 |
| ACRNN (Tao et al. (2020)) | 65.15 ± 8.78 | 64.77 ± 11.19 | 64.02 ± 6.34 | 65.39 ± 8.81 | 66.05 ± 9.51 | 59.96 ± 10.67 | 63.8 ± 4.48 | 55.88 ± 6.81 |
| MTGNN (Wu et al. (2020)) | 67.46 ± 11.51 | 63.03 ± 12.19 | 64.77 ± 7.98 | 67.24 ± 8.33 | 66.66 ± 9.54 | 66.24 ± 11.5 | 63.35 ± 6.29 | 64.01 ± 9.39 |
| RAINDROP (Zhang et al. (2021)) | 66.06 ± 10.11 | 63.7 ± 12.43 | 65.59 ± 7.38 | 64.29 ± 7.98 | 65.74 ± 8.99 | 62.17 ± 10.82 | 65.85 ± 7.61 | 62.44 ± 8.07 |
| HetEmotionNet (Jia et al. (2021)) | [3]69.16 ± 7.94 | 57.84 ± 12.81 | [3]69.0 ± 5.78 | 62.76 ± 7.62 | [2]69.57 ± 8.83 | 68.22 ± 10.18 | [3]69.37 ± 5.58 | 57.21 ± 7.38 |
| TSception (Ding et al. (2022)) | 67.83 ± 9.66 | [1]75.08 ± 10.72 | 63.83 ± 8.79 | [1]71.86 ± 7.97 | 65.48 ± 8.57 | [1]69.93 ± 11.07 | 64.76 ± 7.45 | 60.08 ± 7.74 |
| MEKT (Zhang & Wu (2020)) | 64.24 ± 10.67 | 61.72 ± 11.71 | 63.24 ± 8.14 | 61.9 ± 7.94 | 64.21 ± 8.43 | 67.26 ± 11.21 | 63.62 ± 7.55 | [1]70.92 ± 6.74 |
| MMDA-VAE (Wang et al. (2022)) | [2]70.39 ± 9.56 | [3]67.78 ± 10.76 | [2]70.16 ± 7.02 | 64.51 ± 12.02 | [3]69.36 ± 8.95 | [3]68.38 ± 9.37 | [2]71.32 ± 4.52 | 68.48 ± 9.47 |
| JTSR (Peng et al. (2022)) | 65.53 ± 8.66 | 57.19 ± 9.84 | 62.75 ± 7.35 | 64.88 ± 8.53 | 66.77 ± 7.94 | 66.7 ± 10.94 | 61.7 ± 4.45 | 66.44 ± 8.42 |
| DCCA (Fei et al. (2022)) | 66.81 ± 7.36 | 67.19 ± 10.19 | 65.15 ± 9.41 | 64.08 ± 7.59 | 67.47 ± 8.01 | 63.21 ± 9.97 | 64.93 ± 4.80 | 64.81 ± 7.33 |
| SST-AGCN-DA (Gu et al. (2023)) | 68.25 ± 8.14 | 67.12 ± 9.76 | 66.94 ± 7.06 | 68.89 ± 7.31 | 67.84 ± 8.56 | [1]69.93 ± 9.82 | 65.78 ± 7.27 | 67.88 ± 7.77 |
| MSADA (She et al. (2023)) | 66.5 ± 7.59 | 63.65 ± 10.71 | 67.11 ± 6.75 | 61.79 ± 9.38 | 64.32 ± 9.30 | 60.0 ± 9.33 | 65.0 ± 4.83 | 65.88 ± 8.26 |
| Our VBH-GNN | [1]73.5 ± 7.22 | [2]71.53 ± 10.86 | [1]71.21 ± 6.41 | [2]71.85 ± 7.38 | [1]70.64 ± 7.74 | [2]69.66 ± 9.51 | [1]73.38 ± 4.21 | [3]69.08 ± 6.98 |

**Models and Hyperparameters.** We compared VBH-GNN with two types of baselines. The first type is non-DA methods that focus on the inference of signal relations, *i.e.*, HetEmotionNet (Jia et al. (2021)). The other type is the DA method, *i.e.*, MEKT (Zhang & Wu (2020)). By comparing these two types of baselines, we hope to demonstrate the ability of VBH-GNN to fully use the multi-modal spatial-temporal relationships to improve the cross-subject ER accuracy. All models are trained and tested in the same experimental environment, where all conditions are kept constant except for the hyperparameters of models. The specific environment and parameter settings of VBH-GNN can be found in Appendix A.4.

## 4.2 COMPARISON WITH PRIOR ART

The test accuracy and F1 score over the target domain are shown in Table 1. Compared to modal fusion methods, VBH-GNN significantly improves performance on both datasets. DGCNN (Song et al. (2018)) and EEGNet (Lawhern et al. (2018)) use CNNs to extract the signal's spatial features. ACRNN (Tao et al. (2020)) and TSception (Ding et al. (2022)) include temporal and spatial relational inference components to extract more abundant features and therefore get a higher accuracy than DGCNN and EEGNet. Compared to them, MM-ResLSTM (Ma et al. (2019)) infers the interactivity between modalities through the combination of residual network and LSTM, effectively utilizes the spatio-temporal relationships of multi-modal signals and therefore achieves better results. Based on MM-ResLSTM, HetEmotionNet, and SST-EmotionNet also employ modality-specific feature extractors to learn the heterogeneity of multi-modal signals, thus obtaining excellent results. MTGNN (Wu et al. (2020)) and RAINDROP (Zhang et al. (2021)) are not suitable for physiological signals such as EEG and ECG, so they all achieve low accuracy. None of the above models can learn general spatio-temporal relationships and thus perform less than VBH-GNN in cross-subject scenarios.

Compared to the existing DA methods, VBH-GNN still demonstrates superior performance. MEKT extracts features in the tangent space of the signal and aligns the features through joint probability distribution optimization. JTSR (Peng et al. (2022)) proposes a joint optimization method to optimize label classification while aligning the feature distribution, and MSADA (She et al. (2023)) weighs the signal while aligning the features to improve the accuracy. These methods cannot infer and utilize the spatio-temporal relationships of multi-modal signals, thus limiting their performance. In contrast, DCCA (Fei et al. (2022)) utilizes different feature extractors to capture the heterogeneity of different modalities and infer the spatial relationships of the modalities, thus achieving better results. SST-AGCN-DA (Gu et al. (2023)) further infers the spatio-temporal relations through spatial and temporal modules and thus achieves better accuracy than DCCA. But SST-AGCN-DA fuses all the modalities directly, as compared to MMDA-VAE, which uses different VAEs for different modalities to extract the features before feature fusion, thus achieving higher accuracy. However, all the above methods do not attempt to align the spatio-temporal relationship of multi-modal signal

Table 2: Results of ablation experiments on both datasets. $-$ means to remove this loss and $\downarrow$ means to use minimal weight for the loss. The table shows the results when the weight is set to 0.1.

| Loss | DEAP | | | | DREAMER | | | |
|---|---|---|---|---|---|---|---|---|
| | Arousal | | Valence | | Arousal | | Valence | |
| | Accuracy | F1 Score | Accuracy | F1 Score | Accuracy | F1 Score | Accuracy | F1 Score |
| $-$ BGI Loss | 41.38 | 42.07 | 42.07 | 41.78 | 40.25 | 42.16 | 42.32 | 41.79 |
| $\downarrow$ BGI Loss | 43.45 | 43.64 | 44.28 | 40.75 | 44.83 | 40.31 | 43.98 | 42.59 |
| $-$ EGT Loss | 61.37 | 63.47 | 61.54 | 64.3 | 61.21 | 62.87 | 60.76 | 62.46 |
| $\downarrow$ EGT Loss | 62.1 | 63.27 | 62.44 | 60.23 | 61.32 | 63.92 | 62.88 | 61.3 |
| ALL Loss | [1]**73.5** | [1]**71.53** | [1]**71.21** | [1]**71.85** | [1]**70.64** | [1]**69.66** | [1]**73.38** | [1]**69.08** |

Table 3: Modality-deficient experiments on two datasets. "$-$" denotes removing one modality; "ALL" indicates that all modalities are retained. The number in () indicates the number of channels.

| Modality | DEAP | | | | Modality | DREAMER | | | |
|---|---|---|---|---|---|---|---|---|---|
| | Arousal | | Valence | | | Arousal | | Valence | |
| | Accuracy | F1 Score | Accuracy | F1 Score | | Accuracy | F1 Score | Accuracy | F1 Score |
| GSR (1) | N.A. | N.A. | N.A. | N.A. | GSR (N.A.) | N.A. | N.A. | N.A. | N.A. |
| EOG (2) | 55.08 | 52.51 | 59.12 | 62.23 | EOG (N.A.) | N.A. | N.A. | N.A. | N.A. |
| EMG (2) | 56.87 | 60.74 | 52.61 | 50.84 | EMG (N.A.) | N.A. | N.A. | N.A. | N.A. |
| ECG (N.A.) | N.A. | N.A. | N.A. | N.A. | ECG (2) | 56.35 | 53.55 | 55.21 | 59.06 |
| $-$EEG (5) | 61.21 | 62.73 | 60.42 | 62.03 | $-$EEG (ECG 2) | 56.35 | 53.55 | 55.21 | 59.06 |
| EEG (32) | 65.7 | 65.06 | 64.88 | 64.57 | EEG (14) | 65.51 | 62.08 | 63.59 | 63.31 |
| ALL (37) | [1]**73.5** | [1]**71.53** | [1]**71.21** | [1]**71.85** | ALL (16) | [1]**70.64** | [1]**69.66** | [1]**73.38** | [1]**69.08** |

features between the domains. The RDA included in VBH-GNN finds a more general cross-domain distribution by aligning domain-invariant multi-modal spatio-temporal relationships while circumventing individual differences of signal features. As a result, VBH-GNN achieves better results on the cross-subject ER task than the above models.

## 4.3 ABLATION STUDIES

We conducted ablation studies to evaluate the effects of two main components: BGI loss and EGT loss. Table 2 showed the results on the two datasets. We compared the effect of the following four scenarios on the model performance: 1) removing the BGI loss, 2) removing the EGT loss, 3) the BGI loss obtaining a smaller weight, and 4) the EGT loss obtaining a smaller weight. The results from the two datasets show that the effects of BGI loss and EGT loss on model performance exhibit similar trends. When the BGI loss is wholly removed or given a smaller weight, VBH-GNN only achieves an accuracy of around $40\%$ on both valence and arousal for two datasets. This suggests that the BGI loss determines whether the model converges or not or whether the model can learn the spatio-temporal relationship distribution of modalities. For EGT loss, its effect on the model is to determine the degree of convergence. This effect is more pronounced in the DEAP dataset due to modalities' more complex spatio-temporal relationships. In summary, EGT can help the model find the equilibrium (saddle) points between BGI and the downstream classification task, and the best accuracy is obtained when the two losses are used together.

## 4.4 MODALITY-DEFICIENT EXPERIMENTS

We conducted modality-deficient experiments to verify the rationality of our motivation for using multi-modal signals. As shown in Table 3, "All" denotes using all modalities, and "$-$EEG" denotes removing EEG signals. The results of all modalities show significant improvement compared to incomplete signals. Individual physiological signals lack sufficient information about spatial-temporal relationships, and therefore all perform worse. Since the EEG has the most significant number of channels and contains the wealthiest spatial-temporal relationships, it is still better than all the other signals. The model showed relatively stable results on the DEAP dataset because the DREAMER dataset has fewer modalities. In summary, we conclude that 1) EEG can provide more practical information than other signals, and 2) VBH-GNN can take advantage of complementarity between multi-modal signals by inferring and utilizing spatial-temporal relationships.

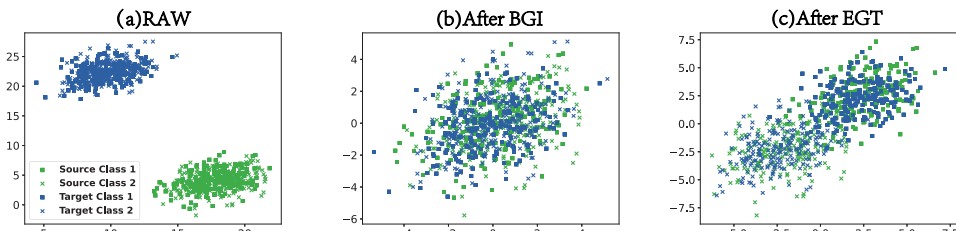

Figure 4: The relationship distribution of source and target domain samples. The left figure is the raw distribution, and the middle and right are the distributions after BGI and EGT procedures.

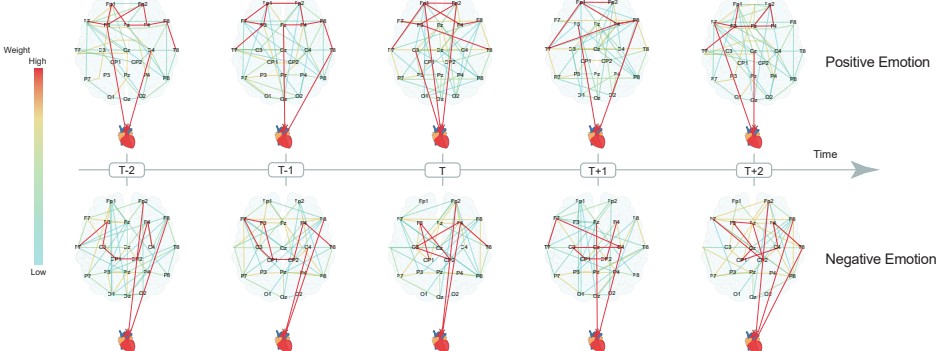

Figure 5: Visualization of the spatio-temporal relationship of the brain inferred by VBH-GNN on DEAP, where $T$ denotes the $T^{th}$ time segment.

## 4.5 DISTRIBUTION VISUALIZATION

We apply T-distributed stochastic neighbor embedding (t-SNE) to visualize the distribution of multi-modal signals. As shown in Figure 4, we show the raw distribution and distribution after BGI and EGT. The raw distributions of the two domains have a low degree of coupling and form a high coupling state after BGI. EGT divides the clustering centers of the two emotion categories in the source and target domains into two. Thus we demonstrate that BGI can align the source and target domains, and EGT can make specific emotion distinctions.

## 4.6 INTERPRETABILITY ANALYSIS

We further verified the interpretability of the relationship after the EGT to visualize the spatio-temporal relationships. As shown in Figure 5 (we remove some neighboring nodes for readability), we demonstrate the spatio-temporal relationships under different emotions. The intracranial relationships are shown as high correlations between the frontal lobe (front part) and frontal cortex (middle front part) under positive emotions; the central sulcus (middle part) was highly correlated under negative emotions, which is consistent with the findings in (Min et al. (2022)). The relationships between modalities are shown by the correlation between the heart and the central sulcus under positive emotions and the correlation between the heart and the right prefrontal cortex (right front part) under negative emotions, which is also consistent with the findings in (Lichtenstein et al. (2008); Kreibig (2010)).

## 5 CONCLUSION

We propose the VBH-GNN for cross-subject ER using multi-modal signals. This is the first time emotional knowledge transfer is achieved by aligning the spatio-temporal relationships of multi-modal signals between domains. The VBH-GNN achieves the inference and alignment of multi-modal spatial-temporal relationships in domains through RDA. Experiment results show that VBH-GNN can improve ER accuracy in cross-subject scenarios and achieve promising results. In addition, the interpretability analysis shows that the distribution of relationships inferred by VBH-GNN is in accordance with the existing physiological findings, which helps reveal the potential working mechanism of VBH-GNN.

## 6 ACKNOWLEDGMENTS

This research is supported by the National Research Foundation (NRF) Singapore, NRF Investigatorship NRF-NRFI06-2020-0001.

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

# A APPENDIX

## A.1 DATASETS

**DEAP** is a multimodal dataset always used to analyze human affective states. There are electroencephalogram (EEG) and peripheral physiological signals collected from 32 participants. Each participant viewed 40 one-minute-long excerpts of music videos and rated them based on arousal, valence, like/dislike, dominance, and familiarity. Besides that, considering the potential correlation between the brain and facial part, frontal face videos were also collected for 22 of the 32 participants. The selection of stimuli was guided by a novel method, which involved retrieving videos based on affective tags from the last.fm website, detecting video highlights, and employing an online assessment tool for evaluation.

In an experiment, 32 volunteers watched a subset of 40 music videos mentioned earlier. During the session, EEG and physiological signals were recorded from each participant, and they also rated the videos based on arousal, valence, like/dislike, dominance, and familiarity. Additionally, frontal face videos were recorded for a portion of the participants. The data set has 40 channels, covering 4 different modes. Channels $1 - 32$ are used to collect participant EEG signals as modality $1$. Channels $33$ and $34$ are used to collect horizontal and vertical EOG signals as modality $2$. As modality $3$, channels $35$ and $36$ are used to collect Zygomaticus EMG and Trapezius EMG signals, respectively. Channel $37$ is used to collect the GSR signal as modality $4$. Channels $38$, $39$, and $40$ collected the Respiration belt signal, Plethysmograph signal, and Temperature signal, respectively, which are not used in our experiences. The data was downsampled to 128Hz.

**Dreamer** is a multi-modal database containing electroencephalogram (EEG) and electrocardiogram (ECG) signals recorded during affect elicitation via audio-visual stimuli. Signals were collected from 23 participants, who self-assessed their affective state after each stimulus regarding valence, arousal, and dominance. The experiment used wearable, portable, and wireless equipment to capture signals, aiming to allow effective computing methods to be available in everyday applications. The Emotiv EPOC wireless EEG headset and the Shimmer2 ECG sensor for ECG were used for EEG.

With film clips, participants were exposed to audio and visual stimuli to evoke emotional reactions, while EEG and ECG data were recorded. The dataset comprised 18 carefully selected film clips intended to elicit emotions. These clips consisted of cut-out scenes from various films that evoke many emotions. Each of the 18 film clips targeted one of the following nine emotions: happiness, amusement, excitement, disgust, anger, fear, sadness, calmness, and surprise. The length of the film clips was between 65 to $393s$ ($M = 199s$)

With the variable duration in the film clips and the allowed time for a specific emotion to become dominant, this approach aimed to capture the most prominent emotional responses elicited by the stimuli, facilitating a more focused and reliable analysis of the EEG and ECG signals.

## A.2 EVALUATION METRICS

**Accuracy** is a metric used to evaluate the performance of classification models, particularly in the context of binary or multiclass classification tasks. It measures the overall correctness of the model's predictions by calculating the proportion of correctly classified samples out of the total samples in the dataset. The definition of Accuracy is as follows: Accuracy = (Number of Correctly Classified Samples) / (Total Number of Samples). In this formula, the Number of Correctly Classified Samples refers to the count of samples for which the model's predictions match the actual accurate labels. Total Number of Samples is the total count of samples in the dataset. Accuracy is typically expressed as a percentage, with values ranging from 0% (worst performance, indicating that the model correctly classified none of the samples) to 100% (best performance, indicating that the model correctly classified all samples). However, it may not be the most informative metric in imbalanced datasets, where one class significantly outnumbers the others. In such cases, other metrics like Precision, Recall, and F1 Score may be more informative for assessing model performance, as they focus on specific aspects of classification accuracy.

**F1-Score** is a metric used to evaluate the performance of binary classification models. It combines two metrics, Precision and Recall, with the goal of providing a balanced assessment of a model's accuracy and completeness. The definition of the F1 Score is as follows: F1 Score = 2 (Precision

$\times$ Recall) / (Precision + Recall). Precision represents the proportion of correctly predicted positive samples out of all predicted positive samples. Precision is calculated as: Precision = (True Positives) / (True Positives + False Positives). Recall represents the proportion of correctly predicted positive samples out of all actual positive samples. Recall is calculated as: Recall = (True Positives) / (True Positives + False Negatives) The F1 Score is a harmonic mean of Precision and Recall, and its values range from 0 to 1, with 0 indicating the worst performance and 1 indicating the best performance. The strength of the F1 Score lies in its ability to provide a balanced assessment of a model's accuracy and capture positive samples, especially when dealing with imbalanced class distributions in the dataset. It is a commonly used performance metric, particularly for evaluating binary classification models, as it balances Precision and Recall and offers a comprehensive understanding of a model's performance.

### A.3 BASELINES

Our baseline consists of two classes: non-DA methods that focus on inferring signal relationships (Song et al. (2018); Lawhern et al. (2018); Ma et al. (2019); Tao et al. (2020); Ding et al. (2022); Wu et al. (2020); Zhang et al. (2021); Jia et al. (2020; 2021)) and DA methods (Zhang & Wu (2020); Wang et al. (2022); Peng et al. (2022); She et al. (2023); Fei et al. (2022); Gu et al. (2023))

**DGCNN** (Song et al. (2018)) is a deep learning model designed for EEG emotion recognition using dynamical graph convolutional neural networks. It constructs graph representations dynamically, conducts graph convolution operations to capture localized relationships, and applies sorting-based pooling to condense graph dimensions and extract pivotal features. Moreover, it acquires global features for diverse applications, including graph classification, node classification, and link prediction. DGCNN exhibits flexibility and adaptability, rendering it apt for various tasks encompassing distinct types of graph structures.

**EEGNet** (Lawhern et al. (2018)) is a compact convolutional neural network for EEG-based BCIs. EEGNet is widely utilized in both BCI applications and neuroscience research. It harnesses 1D convolutions to capture temporal patterns, thus facilitating real-time processing adeptly. EEGNet stands out for its efficacy in tasks such as EEG classification and segmentation, owing to its efficient temporal pattern recognition capabilities.

**MM-ResLSTM** (Ma et al. (2019)) is a sophisticated deep learning architecture crafted for processing data from multiple modalities. MM-ResLSTM amalgamates the strengths of residual connections and Long Short-Term Memory (LSTM) layers. By effectively leveraging these components, MM-ResLSTM adeptly captures relationships and features spanning diverse modalities. This model fosters interactions between modalities, facilitating tasks such as multimodal sentiment analysis, inference, and cross-modal data fusion. By harnessing complementary information from varied data sources, MM-ResLSTM enhances overall performance across a spectrum of applications.

**ACRNN** (Tao et al. (2020)) is an attention-based convolutional recurrent neural network (ACRNN) designed to enhance emotion recognition accuracy by extracting more discriminative features from EEG signals. ACRNN employs a channel-wise attention mechanism to allocate weights to different channels dynamically. Additionally, it utilizes a CNN to capture spatial information encoded within EEG signals. To delve into temporal dynamics, ACRNN integrates extended self-attention mechanisms within a recurrent neural network (RNN). This allows for the recalibration of importance based on intrinsic similarities within EEG signals over time. This holistic approach enables ACRNN to analyze EEG data and thoroughly enhance emotion recognition accuracy.

**TSception** (Ding et al. (2022)) is a multi-scale convolutional neural network that can classify emotions from EEG with its temporal dynamics and spatial asymmetry. Comprising asymmetric spatial, dynamic temporal, and high-level fusion layers, TSception simultaneously learns discriminative representations across time and channel dimensions. The dynamic temporal layer uses multi-scale 1D convolutional kernels to capture temporal and frequency representations in EEG signals. Meanwhile, the asymmetric spatial layer capitalizes on the distinctive asymmetrical EEG patterns associated with emotions, facilitating the acquisition of discriminative global and hemisphere representations. This comprehensive architecture equips TSception with the ability to extract nuanced features from EEG data, enabling robust emotion classification.

**MTGNN** (Wu et al. (2020)) stands out as a specialized framework for multivariate time series inference, distinguishing itself from traditional methods by autonomously learning uni-directional relationships among variables. Its innovative design integrates a mix-hop propagation layer and a dilated inception layer, enabling the capture of both spatial and temporal dependencies within the data. By combining graph convolution, graph learning, and temporal convolution, MTGNN offers a comprehensive approach to modeling complex interactions in multivariate time series data, effectively utilized in fields like energy and traffic.

**RAINDROP** (Zhang et al. (2021)) is a graph neural network designed for analyzing irregularly sampled and multivariate time series data, often found in healthcare, biology, and climate science. It uniquely handles time series data challenges by representing each data point as a separate sensor graph, capturing the dynamic relationships between sensors over time. RAINDROP excels in estimating the underlying structure of these sensor networks and uses this knowledge to make accurate predictions. It outperforms existing methods significantly, even in complex scenarios where sensors are omitted, proving especially effective in classifying time series data and understanding temporal patterns.

**SST-EmotionNet** (Jia et al. (2020)) represents a groundbreaking approach to emotion recognition, leveraging a spatial-spectral-temporal (SST) paradigm within a 3D dense network architecture. Its primary strength is seamlessly integrating spatial, spectral, and temporal features within a unified network framework. Additionally, it incorporates a 3D attention mechanism to explore discriminative local patterns dynamically.

**HetEmotionNet** (Jia et al. (2021)) is a pioneering two-stream heterogeneous graph neural network engineered to fuse multi-modal physiological signals for enhanced emotion recognition. It comprises both the spatial-temporal and the spatial-spectral stream, adept at integrating spatial-spectral-temporal domain features within a cohesive framework. Each stream is carefully designed, incorporating the graph transformer network to handle heterogeneity, the graph convolutional network to capture correlations, and the gated recurrent unit for capturing domain dependency.

**MEKT** (Zhang & Wu (2020)) offers a novel method for transferring knowledge, particularly in EEG trial analysis. Initially, MEKT aligns the covariance matrices of EEG trials within the Riemannian manifold, effectively capturing underlying geometric structures. Between the source and target domains, it extracts features within the tangent space, optimizing domain adaptation by minimizing the joint probability distribution change. MEKT ensures that the geometric structures inherent to both domains are preserved while adapting. Remarkably, MEKT is versatile, capable of managing single or multiple source domains, and is computationally efficient, making it suitable for practical applications in EEG data analysis.

**MMDA-VAE** (Wang et al. (2022)) presents a novel approach to learning shared cross-domain latent representations of multi-modal data. This method employs a multi-modal variational autoencoder (MVAE) to build data from modalities into a unified space. By integrating adversarial learning and cycle consistency regularization, the method effectively minimizes distribution discrepancies among different domains within the shared latent representation layer. As a result, it facilitates knowledge transfer across domains, enabling the extraction of meaningful insights from multi-modal data while preserving domain-specific characteristics.

**JTSR** (Peng et al. (2022)) optimizes the shared subspace projection matrix and target labels simultaneously. By jointly optimizing these components, the model achieves superior performance in emotion recognition tasks. Furthermore, it identifies spatial-frequency activation patterns within critical EEG frequency bands and brain parts associated with cross-subject emotion expression. This analysis is facilitated by examining the learned shared subspace, enabling a quantitative understanding of the underlying patterns that contribute to emotion recognition enhancements.

**MSADA** (She et al. (2023)) utilizes a multisource associate domain adaptation (DA) network, accounting for domain-specific and domain-invariant features. Initially, distinct parts are established for multiple resource domains, assuming shared low-level features among EEG datasets. Subsequently, domain-specific features are extracted employing the one-to-one associate DA approach. Weighted scores of specific sources are determined based on distribution distance, enabling the derivation of different source classifiers with exact weighted scores. Finally, EEG emotion recognition tests are conducted across different datasets.

**DCCA** (Fei et al. (2022)), based on Deep Canonical Correlation Analysis (CDCCA), works by transforming each modality separately and coordinating the multi-modals into a hyperspace using specific canonical correlation analysis constraints. In the proposed test, eye movement signals are the sole inputs used during the test phase, while the knowledge derived from EEG signals is integrated during the training stage. The model processes these eye movement signals, allowing it to infer human decision confidence levels, thus enabling the application of eye movement signals in scenarios traditionally reliant on more complex and costly EEG signal processing.

**SST-AGCN-DA** (Gu et al. (2023)) is a deep learning model designed for cross-subject decision confidence estimation. Unlike the traditional SST-AGCN model, which is focused on specific subjects and may exhibit decreased efficiency in cross-subject scenarios, the SST-AGCN-DA is tailored for more common application scenarios involving various subjects. The model incorporates domain adaptation techniques to enhance its applicability and accuracy across different subjects. It demonstrates superior performance in experimental comparisons, particularly in cross-subject confidence estimation experiments conducted on an EEG dataset during a text-based decision-making experiment.

## A.4 IMPLEMENTATION DETAILS

For our experiments, we utilized a computational environment fortified with specific software and hardware configurations. On the software front, our setup was rooted in Python version 3.8.11, complemented by PyTorch (version 1.8.0) and NumPy (version 1.20.2). Hardware-wise, our system was powered by an Intel (R) Xeon (R) CPU E5-2620 v4 clocked at 2.10 GHz, buttressed by a substantial 256 GB of RAM, and accelerated by a GeForce Tesla P40 GPU. As for the configuration of the training hyper-parameters, the details are listed in Table 4.

| Hyper-parameter | Value |
|---|---|
| HiddenDim | 128 |
| GraphDim | 128 |
| Sequence Dim | $4 \times$ sample rate |
| Node Embedding Dim | 128 |
| Edge Embedding Dim | 128 |
| Epoch | 500 |
| Time Series | 32 |
| Weight Decay | $1e-3$ |
| Learning Rate | $3e-5$ |
| eps ($\epsilon$) | $1e-5$ |
| Dropout | 0.5 |
| $f_{emb}$ Conv1 | $0.1 \times$ sample rate |
| $f_{emb}$ Conv1 Channel | 8 |

Table 4: The Configuration of Hyper-parameters for Training.

