# OpenReview forum: "VBH-GNN: Variational Bayesian Heterogeneous Graph Neural Networks for Cross-subject Emotion Recognition"
_ICLR.cc/2024/Conference — ICLR 2024 poster_

### Official Review · Reviewer_zn1U · 2023-10-27

**Soundness:** 3 good
**Presentation:** 3 good
**Contribution:** 3 good
**Rating:** 8
**Confidence:** 4

**Summary:**

The authors propose a Domain adaptation method for a cross-subject emotion recognition task by aligning spatial-temporal relationships of multimodal physiological signals. The method is implemented as follows: the model contains temporal and spatial Relationship Distribution Adaptation (RDA) components, each of which represents the multimodal spatio-temporal relationships as edge distributions of a heterogeneous graph and aligns them twice.

**Strengths:**

Exploiting the correlation of multimodal physiological signals to address individual differences in cross-subject emotion recognition is a simple but often overlooked detail in the past. Multimodal data has been shown to provide more information compared to single modalities, and making full use of multimodal data can improve model performance. In other words, it is a common approach to utilise the complementary of multimodal data for feature fusion to generate better feature representations. However, VBH-GNN adopts a different perspective, i.e., exploiting the relationship between multimodal data to solve the problem of individualised differences. This is like in NLP, different languages may be completely different in pronunciation and writing, but there must be similar correlations in semantic structure. This multimodal relationship will be better represented in the field of physiology, because various physiological signals correspond to the physiological system interactions of the human body. In the final experimental section (4.6) the authors also provide a good demonstration of the hypothesis that there is cross-subject similarity in the inter-signal relationships, despite individual differences in signals.

Domain alignment via heterogeneous graph edge distributions is a novel idea in the field of BCI.DA has been used relatively rarely in EEG tasks, but most of the concerns are about finding domain invariant features. These features are often signal-level features. The authors argue that domain invariant features tend to trap the model in sub-optimal solutions due to individual differences in physiological signals. Therefore VBH-GNN looks for domain-invariant distributions (although in a sense the spatio-temporal relationship distribution is also a signal feature, I think there is a difference between the feature representation and the distribution), which has not been seen in previous EEG tasks.

Bayesian graph inference (BGI) is a generalised method. It implements a distribution with infinite parameter n in a neural network by an ingenious method and backpropagation of this infinite parameter n in network by an upper bound. It provides a feasible method to implement infinite parameters in neural networks as well as to perform backpropagation.

**Weaknesses:**

The explanation of the (Emotional graph transform) EGT step lacks depth. Although the authors demonstrate in their experiment (4.5) the difference between EGT and BGI, which is able to transform an intermodal heterogeneous graph into a more emotion-specific graph, the motivation for this step is not clear enough to me, and it seems more like a step based on experimental attempts to determine what to do; in other words, the authors seem to know what has to be done and how it should be done, but are unable to explain why it allows HetG to be transformed in an emotionally weighted way. what has to be done and how it should be done, but are unable to explain why doing so allows HetG to undergo an emotionally relevant weighting transformation. I think it should be that EGT creates an Attention-like effect between the original input and the HetG weights, and in training this ATTENTION tends to notice the HetG edges that are more emotionally relevant. I think the authors should experimentally demonstrate what makes EGT work and provide a more direct explanation in the paper.

Lack of a flowchart of the overall model. This paper contains a large number of formulas that are difficult to read, and coupled with the lack of a flowchart of the overarching model, I had a hard time imagining what the complete model would look like, how the temporal and spatial RDA components would be linked, and how the heterogeneous edges would be generated. Although the authors used formulas to explain the steps, this piling up of formulas in the presence of a large number of formulas rather made it difficult for me to understand the framework of the model, at least for me I would have liked a clearer flowchart as a guide.

The choice of Baseline is rather narrow. Although several baselines are included, they are basically methods in the BCI area. There are many DA methods in other fields, such as Maximum Classifier Discrepancy proposed in CVPR and a series of methods derived from it. I think adding more diversity of DA methods to compare and analyse can make the results more convincing, as DA is a relatively uncommon method in the BCI domain. Because multimodal data in other area do not necessarily correlate across subject as well as physiological signals, so comparison with methods in other area can demonstrate the applicability of VBH-GNN in the field of physiological signals.

**Questions:**

Is the BCI in Figure 2 trying to represent BGI?

---

> ### Author Response · Authors · 2023-11-19
> **Response to Reviewer zn1U (1/2)**
>
> Thank you so much for the positive rating and insightful comments. Your valuable suggestions are beneficial for further strengthening our paper. We have revised our paper according to your comments.
>
> ## Answer to Weakness 1:
>
> If we understand the reviewers correctly, the reviewer believes that our explanation of what and how EGT plays a role in RDA was unclear. Thanks to the reviewers' suggestions, we have updated the description of the EGT section. Here, we would like to clarify the role of EGT in terms of its rationale.
>
> **The role of EGT is to distinguish between the latent relationship distribution founded by BGI in different emotions**. We have mentioned it in our paper:
>
> > EGT divides the clustering centers of the two emotion categories in the source and target domains into two.
>
> Therefore, the EGT is a component designed for downstream ER tasks for extracting emotion-related information representations from HetG (output of BGI). As shown in our paper in Fig.4（c）, the distribution after EGT can be more adapted to the downstream classification task. As shown in Section 4.3 ABLATION EXPERIMENTS, there is a significant decrease in the accuracy after the EGT Loss is removed. We have summarized the effect of EGT on VBH-GNN as follows:
>
> > For EGT loss, its effect on the model is to determine the degree of convergence.
>
> In other words, the EGT can be regarded as a bridge between downstream tasks and BGI. It makes the domain-invariant relationship distribution inferred from BGI more suitable for downstream tasks.
>
> **The EGT is achieved by transforming the HetG by weighting each edge with a conditional variable**. This conditional variable is a Gaussian distribution computed from edge embedding and conditioned on the edges of HetG:
>
> > $\mathcal{N} _{s\lor lt}|Z _{\text{HetG}} \sim \mathcal{N}(Z _{\text{HetG}} \times \bar{\mu} _{s\lor lt}, Z _{\text{HetG}} \times \bar{\sigma }^2 _{s\lor lt})$
>
> It integrates the emotion information from node embedding into the graph structure. Therefore, we use the re-parameterization trick on this conditional variable to make it a weight related to the emotion, which will be applied further to transform the HetG to EmoG.
>
> ## Answer to Weakness 2:
>
> According to the reviewer's suggestion, **we have added a flowchart of the VBH-GNN in our paper (Please check Figure 2 in our revised PDF version)**. We have shown the workflow of the VBH-GNN and the critical components. We also added a more detailed explanation of each component in Section 3.1. We hope this will help the reader to understand VBH-GNN better.

---

> ### Author Response · Authors · 2023-11-19
> **Response to Reviewer zn1U (2/2)**
>
> ## Answer to Weakness 3:
>
> We apologize for not clearly stating our criteria for baseline selection. We and the reviewers have some different perspectives on baseline selection, so we wanted to share our opinions. We strongly agree with the reviewers and have added new experiments to broaden the scope of our baseline.
>
> **First, the physiological signal is a time series signal, and the MCD series method is unsuitable for this scenario**. The MCD series of methods are more likely to perform tasks such as object classification. Compared with this type of data, physiological signals have a significant feature: time correlation. Therefore, the model needs to analyze the spatial-temporal relationship of the data. Due to different data scales, these models are difficult to transplant to cross-subject ER tasks directly.
>
> **Second, to increase baseline diversity, we added two baselines from other fields**. Baseline in [1] and [2] apply graphs to learn spatial relationships between multivariate time series data. This kind of data is consistent with the multimodal physiological signals we use. We have updated the experimental results in Table 1:
>
> | Method | DEAP |  |  |  | DREAMER |  |  |  |
> | :---: | :---: | :---: | :---: | :---: | :---: | :---: | :---: | :---: |
> |  | Arousal |Arousal  | Valence | Valence | Arousal | Arousal | Valence |Valence   |
> |  | Accuracy | F1 Score | Accuracy | F1 Score | Accuracy | F1 Score | Accuracy | F1 Score |
> | MTGNN [8] | $67.46 \pm 11.51$ | $63.03 \pm 12.19$ | $64.77 \pm 7.98$ | $67.24 \pm 8.33$ | $66.66 \pm 9.54$ | $66.24 \pm 11.5$ | $63.35 \pm 6.29$ | $64.01 \pm 9.39$ |
> | RAINDROP [9] | $66.06 \pm 10.11$ | $63.7 \pm 12.43$ | $65.59 \pm 7.38$ | $64.29 \pm 7.98$ | $65.74 \pm 8.99$ | $62.17 \pm 10.82$ | $65.85 \pm 7.61$ | $62.44 \pm 8.07$ |
> | Our VBH-GNN | **73.5** $\pm$ **7.22** | **71.53** $\pm$ **10.86** | **71.21** $\pm$ **6.41** | **71.85** $\pm$ **7.38** | **70.64** $\pm$ **7.74** | **69.66** $\pm$ **9.51** | **73.38** $\pm$ **4.21** | **69.08** $\pm$ **6.98** |
>
> However, these models do not perform well because the data in their applicable scenarios differ from physiological signals regarding the sparsity and sampling rate, etc.
>
> [1] Zonghan Wu, et al. "Connecting the dots: Multivariate time series forecasting with graph neural networks." ACM SIGKDD, 2020
>
> [2] Xiang Zhang, et al. "Graph-Guided Network for Irregularly Sampled Multivariate Time Series." ICLR, 2022.
>
>
>
> ## Answer to Question 1:
>
> We thank the reviewer for carefully reading our manuscript and pointing out this typo. **We examined the manuscript thoroughly and tried our best to correct all the typos we found**. We also have redrawn Figure 2 (Now it is Figure 3 in the current version). We hope this could provide readers with a better reading experience.

---

### Official Review · Reviewer_D38r · 2023-10-31

**Soundness:** 3 good
**Presentation:** 3 good
**Contribution:** 3 good
**Rating:** 8
**Confidence:** 4

**Summary:**

The paper addresses cross-subject emotion recognition using EEG data. It proposes a Variational Bayesian Heterogeneous Graph Neural Network (VBH-GNN) with Relationship Distribution Adaptation (RDA) to align spatio-temporal relationships between multi-modal physiological signals, instead of aligning individual features. The method outperforms existing approaches in experiments on two public datasets.

**Strengths:**

- The authors come up with an interesting method to combat heterogeneity across patients by using multiple modalities.
- The authors provide a rigorous proof and interpretability of their method.

**Weaknesses:**

- In Table 1, there is a typo. I believe DEAT should read DEAP. Additionally, Mathod should be Method. Additionally, expanding Table 1 by including which modalities were used for each baseline would be more comprehensive.
- Expanding Table 3 by seeing the effect of all modalities used would be more comprehensive.

**Questions:**

- The paper assumes an infinite number of edges between nodes with probabilities $p_n​$ tending to zero in section 3.2.1. Could the authors clarify a bit more on the reasoning behind the assumption?
- Could the authors clarify $\epsilon$ in equation 12 in the main paper. They mention it is a "very small hyperparameter." Does this mean that it is close to 0?
- Did the authors confirm that for Table 1, the comparable methods are all using the same experimental conditions as the authors (e.g., cropped experiment, experiment setup, modalities used).

---

> ### Author Response · Authors · 2023-11-19
> **Response to Reviewer D38r (1/2)**
>
> We sincerely thank the reviewer for the positive rating and insightful comments. Your valuable suggestions are beneficial for further strengthening our paper. We have revised our paper according to your advice.
>
> ## Answer to Weakness 1:
> Thank you for carefully reading our manuscript and pointing out the typos. **We have revised these two typos**.
>
> Following the reviewer's suggestion, **we have added the modality information in Table 1**. The Deap dataset contains EEG, EOG, EMG, and GSR modalities, and the Dreamer dataset contains EEG and ECG modalities. Since our method and baselines all utilize the full modalities available in each dataset, we do not specify the modalities used for individual baselines due to space constraints. Instead, we annotate the modalities included in each dataset in the table headers (see Table 1 on Page 8). Besides we have also added the description of modality information in the experiment setting section (see 4.1 EXPERIMENTAL SETTING).
>
>
> ## Answer to Weakness 2:
> According to the reviewer's suggestion, **we have substantially expanded Table 3 to include experimental results for all individual and combined modalities**. The updated Table 3 with these additional results can be found on page 8 of the revised PDF paper.

---

> ### Author Response · Authors · 2023-11-19
> **Response to Reviewer D38r (2/2)**
>
> ## Answer to Question 1:
> We will explain this hypothesis through the physiological phenomena behind it as well as similar examples from other fields.
>
> **The explanation of the infinity edges**: Physiological systems are highly complex and contain many biological processes that interact with each other in multiple ways. While some of these interactions are medically proven, such as heart rhythms altering skin temperature and unconscious eye movements due to brain activity, there are more complex and yet-to-be-fully explored interrelationships [3]. We, therefore, use edges that tend to be infinite to represent the complexity and diversity of signal relationships in physiological systems.
>
> **The explanation of the probability of existence for edges close to 0**: Physiological systems often have multiple regulatory mechanisms, and the relationships between physiological signals may be multilevel and sublinear. Thus, most relationships between physiological signals are insignificant, and only a few relationships are meaningful in specific situations (e.g., emotions). Therefore, to reflect the sparsity of the relationships between physiological signals, we assumed that the probability of the existence of edges tends to 0 before determining the pattern of emotions.
>
> **Similar examples**：Similar assumptions are applied to the natural language field. For example, as mentioned in [1] and [2], there are relational structures between consecutive utterances, such as comments, analogies, and contrasts. Before producing complex relational structures among utterances, the listener unconsciously generates an infinite amount of similar data structures while the probability of relation existence is minimal (close to 0).
>
>
>
> ## Answer to Question 2:
>
> The $\epsilon$ is a hyperparameter with a very small value 1e-5. **The $\epsilon$ had been listed in Table 5 in Appendix (A.5) of our original submission, but we inappropriately used a different symbol**.
> We have updated the $\epsilon$ in Table 5 in the Appendix (A.5), and we have also added the description of $\epsilon$ in our revised submission for better clarification (see Line xxx).
>
> | Hyper-parameter | Value |
> | :--- | :---: |
> | $\vdots$ | $\vdots$ |
> | eps ($\epsilon$) | $1 \mathbf{e}-5$ |
> | $\vdots$ | $\vdots$ |
>
> ## Answer to Question 3:
>
> Yes, all the comparable methods use the same experimental conditions(i.e., cropped experiment, training strategy, and modalities). We have a shared Dataloader for all models.
> Here are some more specific descriptions of our experiment settings:
>
> **Modalities:**
> In Table 1, **all baselines use the same modalities**: the EEG, EOG, EMG, and GSR modalities from DEAP and the EEG and ECG modalities from DREAMER.
>
> **Training strategy:**
> **We adopt the "leave-one-subject-out" (LOSO) paradigm to divide the dataset and implement a shared Dataloader**. We use one subject as the target domain and the remaining as the source domain. The training set consists of all source domain and some target domain data, and the remaining target domain data are divided into validation and testing sets.
>
> **Cropping Experiment:**
> **We employed a specific data cropping strategy for each trial**, segmenting the data into 4-second non-overlapping intervals for all methods. This data segmentation was conducted only after dividing the dataset into training and testing sets. Given the inherent time-series nature of physiological signals, adjacent segments are naturally correlated. If these segments were distributed before splitting training and testing sets, it could lead to data leakage, inflating model performance metrics like accuracy and F1 score. To ensure the reliability of our results, we applied the same cropping strategy across all models. This approach helps prevent abnormally high performance resulting from data leakage, ensuring a fair and accurate evaluation of model efficacy.
>
>
> ## Reference
> [1] Murai, Yuki, and Yuko Yotsumoto. "Optimal multisensory integration leads to optimal time estimation." Scientific reports 8.1 (2018): 13068.
>
> [2] Alexander, Patricia A. "Relational thinking and relational reasoning: harnessing the power of patterning." NPJ science of learning 1.1 (2016): 1-7.
>
> [3] Andreassi, John L. Psychophysiology: Human behavior and physiological response. Psychology press, (2010).

---

> ### Comment · Reviewer_D38r · 2023-11-22
> **Response to Rebuttal**
>
> Hello Authors,
>
> Thank you for the clear explanations and expansions to existing tables and experiments.
> I have raised my initial score of 6 to a 8.

---

> > ### Author Response · Authors · 2023-11-23
> >
> > We sincerely thank the reviewer for taking the time to provide feedback and raise the score. We are very pleased that our response addressed the reviewer's concerns.

---

### Official Review · Reviewer_Pxr2 · 2023-10-31

**Soundness:** 3 good
**Presentation:** 3 good
**Contribution:** 3 good
**Rating:** 6
**Confidence:** 3

**Summary:**

This paper designs a Variational Bayesian Heterogeneous Graph Neural Network (VBH-GNN) with Relationship Distribution Adaptation (RDA) for cross-subject emotion recognition (ER) that does not align the distribution of signal features but rather the distribution of spatio-temporal relationships between features. Extensive experiments demonstrate the superiority of the method.

**Strengths:**

1. This method is novel and intuitive.
2. The experiments have clearly demonstrated the effectiveness of the proposed method.

**Weaknesses:**

1. The quality Figure 2 needs to be improved.
2. Statistical results in tables will be more convincing. In addition, the optimal results in Table 1 should all be highlighted.

**Questions:**

1. What is the difference between Spatial-RDA and Temporal-RDA, and what roles do they play in this task?
2. Are the weights of each loss function in Formula 5 the same? Are the weights of each loss function considered?

---

> ### Author Response · Authors · 2023-11-19
> **Response to Reviewer Pxr2**
>
> Thank you so much for your thoughtful comments and the time to provide constructive feedback to strengthen our work further! We have revised the paper to address your concerns as follows.
>
> ## Answer for Weakness 1:
> **We have redrawn Figure 3 (i.e., Figure 2 in the original version) to improve the readability of the RDA module**. Specifically, we use different background colors to denote different sub-modules and thus make the figure details more transparent.
>
>
> ## Answer for Weakness 2:
>
> In the tables of our original submission, we only report two types of statistical results, i.e., Accuracy and F1 Score. Following the reviewer's suggestion, we have added a new type of statistical results in Table 1. **We have added the standard deviation of all Accuracy and F1 Score**. Please see Table 1 in our revised PDF version. We have been conducting experiments to calculate the standard deviation of accuracy results for other tables.
>
> To increase the readability of the results in Table 1, we have highlighted the best, second-best, and third-best results in red, green, and blue, respectively.
>
>
> ## Answer for Question 1:
>
> **Structurally, Spatial RDA and Temporal RDA are the same; functionally, they infer and align relationship distributions of domains in the temporal and spatial dimensions, respectively**. Physiological signals are often interrelated in both time and space. Therefore, to fully infer and utilize this spatio-temporal relationship, we used a Spatial RDA and a Temporal RDA for the multi-modal physiological signals in different stages, respectively.
>
> We have added a global flow chart in Figure 2 to illustrate the role of each module (e.g., Spatial RDA and Temporal RDA). We have also updated the description of the two RDAs in our paper as follows:
>
> > RDA, as the core component of VBH-GNN, accepts node embeddings for domain alignment and updates the weights of node embeddings. The VBH-GNN contains structurally consistent Temporal RDA and Spatial RDA, which perform inference and alignment of relationship distributions in the temporal and spatial dimensions. The details of RDA will be explained in Section 3.2.
>
>
> ## Answer for Question 2:
> **Yes, the weights of all the loss functions in Formula 5 are set to be the same; specifically, the weights are equal to one**.
>
> We have conducted additional experiments to find the best weight combination. We found that using a weight of 1 for all terms achieves the best experimental results. Therefore, we omitted the weights symbolic in the previous submission version. Based on the reviewer's comment, we have added the weight symbols in Formula 5 for a better understanding. The revised Formula 5 is as follows:
>
> > **Loss of VBH-GNN** contains two types of loss: the RDA Loss for aligning the source domain and target domain and the prediction loss of the classifier. The final loss function is formulated as
> > $${\cal L} _{\text{VBH-GNN}} = \lambda _{1}{\cal L} _{\text{SRDA}}+\lambda _{2}{\cal L} _{\text{TRDA}}+ \lambda _{3}{\cal L} _{SBCE}+\lambda _{4}{\cal L} _{TBCE}$$ where ${\cal L} _{\text{SRDA}}$ and ${\cal L} _{\text{TRDA}}$ are loss of Spatial RDA and Temporal RDA (will be further explained in Section 3.2.3), ${\cal L} _{SBCE}$ and ${\cal L} _{TBCE}$ are Binary Cross-Entropy Loss for source and target domain classification. $\lambda _{1}$, $\lambda _{2}$, $\lambda _{3}$, and $\lambda _{4}$ are loss weights, which are all set to $1$ in the experiments.
>
> We hope that our answers clarify your concerns. Thank you for your time and feedback! We are glad to have any further discussion.

---

### Official Review · Reviewer_QvEM · 2023-11-14

**Soundness:** 3 good
**Presentation:** 3 good
**Contribution:** 2 fair
**Rating:** 6
**Confidence:** 2

**Summary:**

This paper discusses the emerging field of research on human emotion using electroencephalogram (EEG) data, with a focus on cross-subject emotion recognition (ER). The challenges in this area include the neglect of multi-modal physiological signals and the difficulty in matching signal features across different domains. To address these issues, the authors propose a novel approach called Variational Bayesian Heterogeneous Graph Neural Network (VBH-GNN) with Relationship Distribution Adaptation (RDA). This method does not align the distribution of signal features but rather focuses on the distribution of spatio-temporal relationships between features. Through extensive experiments on DEAP and Dreamer datasets, the VBH-GNN with RDA demonstrates superior performance compared to state-of-the-art methods.

**Strengths:**

1. This article offers an in-depth analysis of the current challenges within the emerging domain of human emotion recognition using electroencephalogram (EEG), with a specific emphasis on cross-subject emotion recognition (ER).
2. Author introduces the novel VBH-GNN method, and conducts comprehensive experiments to showcase the model's competitive performance.

**Weaknesses:**

1. It is essential to provide a more detailed explanation of how each component contributes to addressing the proposed issue, e.g., clarify the necessity of graph neural network and the specific problem it aims to solve.
2. The readability and visual appeal of the process diagram for RDA in Figure 2 could be improved.
3. The experimental setting of dividing source and target domains is unrealistic.

**Questions:**

1. After constructing the emotional graph, did the author only perform a single convolution layer by multiplying an adjacency matrix with node embeddings? Did the author consider using multiple convolution layers or using the more expressive graph neural network?
2. Due to the necessity of employing Bayesian graph inference to construct multiple graphs in the author's method, I am concerned about whether the performance benefits outweigh the increased computational burden.
3. In Table 1, VBH-GNN achieves optimal accuracy, but the F1 scores consistently demonstrate poor performance. Does this observation imply the class imbalance issue in the predicted results of the method?
4. The authors used leave-one-subject-out paradigm to divide the source and target domains is unrealistic. In this case, the target domain is actually a validation set, which is not the real domain adaptation setting.
5. The heterogeneity (e.g., EEG, ECG) in this paper can be regarded as multi-variant time series data. Please clarify the difference between the heterogeneity and multi-variant time series. The experiments should also include the baseline methods for learning multi-variant time series (e.g., learning a graph structure to represent the spatio relationships in multi-variant time series).

---

> ### Author Response · Authors · 2023-11-19
> **Response to Reviewer QvEM (1/4)**
>
> We sincerely thank the reviewer for the insightful and valuable comments! We have addressed the points raised by the reviewer in detail below and updated our paper based on your comments. The response is a bit late since we have supplemented some experiments during the rebuttal period. Please accept our apologies. Please do not hesitate to post additional comments and questions; we will be more than happy to address them.
>
> ## Answer to Weakness 1:
>
> If we understand correctly, the reviewer means that our explanation of several essential components (e.g., BGI, EGT) in RDA is not detailed enough, which causes difficulties in understanding. Thank you for pointing out this weakness, and **we have updated our paper with more detailed descriptions for each module of RDA**. Here, we explain its core components.
>
> **The Bayesian Graph Inference (BGI) ensures that the model can find the latent relationship distribution of multi-modal physiological signals shared by the source subjects and the target subject**. As we analyzed in Section 4.3:
>
> > BGI loss determines whether the model converges or not or whether the model can learn the spatio-temporal relationship distribution of modalities.
>
> When the BGI is removed, the model fails to converge due to its inability to learn this latent relationship distribution.
>
>
> **The Emotional Graph Transform (EGT) ensures that the model can distinguish between the latent relationship distribution of multi-modal physiological signals in different emotions**. In Section 4.3:
> > For EGT loss, its effect on the model is to determine the degree of convergence.
>
> When the EGT is removed, there is a significant decrease in classification accuracy, and thus, it determines how well the model converges on the ER task.
>
> **We have also added a flow chart of our method VBH-GNN to illustrate different modules.** We graphically show the different modules, including Wav-to-Node, Spatial RDA, Temporal RDA, and Classifier, and also update their function description in Section 3.1.
>
> ## Answer to Weakness 2:
>
> We have redrawn Figure 3 (i.e., Figure 2 in the original version) to improve the readability of the RDA module. Specifically, we use different background colors to denote different sub-modules and thus make the figure details more transparent. We have also added a global flow chart in Figure 2 to illustrate the role of each module in the overall process. Please check Figure 2 and Figure 3 in our revised PDF version.

---

> ### Author Response · Authors · 2023-11-19
> **Response to Reviewer QvEM (2/4)**
>
> ## Answer to Weakness 3 and Question 4:
>
> We used the leave-one-subject-out (LOSO) paradigm to divide source and target domains. If we understand correctly, the reviewer means that the validation and testing sets should originate from different domains (subjects) and should not overlap. We explain why we use LOSO as follows.
>
> **First, there are differences between domain adaptation (DA) and domain generalization (DG).** Our paper aims at the DA problem in a cross-subject emotion recognition (ER) task. The DA is similar to DG, and we speculate whether this might be causing the misunderstanding.
>
> The main difference between DG and DA is whether the model can use the data from the target domain in the training process. For the DG, the subjects in the testing set are regarded as the target domain, and these subjects should be unseen to the trained model. To evaluate the trained model, the validation set should also be from the target domain but use a different subject from the testing set. In this case, using LOSO is unrealistic. However, for the DA, the training set typically contains data from both the source and target domains. Without access to target domain data, models cannot appropriately adapt to the new unknown domain during initial training. In other words, the DA implies using data from source subjects and the target subject for ER, so the LOSO can be used to select the target subject.
>
>
> **Second, the LOSO is a commonly used experiment setup for domain adaptation (DA) in EEG-based cross-subject ER**. To support our point, we cite several typical papers and quote their relevant contents as follows:
>
> 1. Mentioned in ref [1]:
> > Specifically, only one experiment from each subject is involved in the **leave-one-subject-out** cross strategy to study the inter-subject variability.
>
> 2. Mentioned in ref [2]:
> > Based on the above EEG dataset, we adopt the **leave-one-out-cross-validation** method in the following experiments to evaluate the performance of the models in cross-subject scenarios.
>
> 3. Mentioned in ref [3]:
> > On the control experiments, we employ two transfer paradigms, i.e., 'one-to-one' and **'multi-to-one'**. ... . In the latter paradigm, when **one subject serves as the target, all the remaining subjects form the source**.
>
> Despite their different paradigm names, they all randomly select one subject as the target domain and the rest of the subjects as the source domain. We follow their settings in our experiments.
>
> **Third, the division of the training, testing and validation sets in our experiments is also common, and there is no data leakage**. We will explain this with the content of a published paper.
>
> Mentioned in the baseline MMDA-VAE [4]:
>
> > **The training set consisted of the source data and the labelled target data**, i.e., all the samples from the source session and samples from the first three or four trials (one trial per class) in the other target session.
>
> This means that the training set contains source and target domain data.
>
> > We used **samples from the second three or four trials in the target session as the validation set**.
>
> This indicates that the validation set comprises part of the target domain data.
>
> > The samples from **the rest of the twelve or sixteen trials in the target session were used to evaluate classification accuracy**.
>
> This suggests that the testing set is also derived from the target domain but does not intersect with the training and validation sets. This experiment setting is the same as ours, where part of the target domain is included in the training set, while the training, validation, and testing sets do not overlap.
>
> **In summary, our experiment setting (i.e., LOSO paradigm) is commonly used in the cross-subject ER field and does not result in data leakage**. At the same time, we agree with the experiment setting that the reviewer suggested, and we will take the relatively more challenging DG task as our further research direction. We look forward to discussing this with you and thank you for noting the details of the experiment setup.
>
> [1] Zhao, Li-Ming, Xu Yan, and Bao-Liang Lu. "Plug-and-play domain adaptation for cross-subject EEG-based ER." Proceedings of the AAAI Conference on Artificial Intelligence. Vol. 35. No. 1. 2021.
>
> [2] Gu, Rong-Fei, et al. "Cross-Subject Decision Confidence Estimation from EEG Signals Using Spectral-Spatial-Temporal Adaptive GCN with Domain Adaptation." 2023 International Joint Conference on Neural Networks (IJCNN). IEEE, 2023.
>
> [3] Peng, Yong, et al. "Joint feature adaptation and graph adaptive label propagation for cross-subject ER from EEG signals." IEEE Transactions on Affective Computing 13.4 (2022): 1941-1958.
>
> [4] Wang, Yixin, et al. "Multi-modal domain adaptation variational autoencoder for eeg-based ER." IEEE/CAA Journal of Automatica Sinica 9.9 (2022): 1612-1626.

---

> ### Author Response · Authors · 2023-11-19
> **Response to Reviewer QvEM (3/4)**
>
> ## Answer to Question 1:
>
> If we understand the reviewer correctly, the reviewer asks whether we considered using a more expressive network layer to replace the RDA's Graph Attention (GA) module. We have tried using Graph Convolutional Network layers (GCNs) instead of GA, but it is unnecessary. The reasons are as follows.
>
> **The output of EGT (EmoG) already adequately represents the relationships between nodes, so it does not need more expressive networks (such as GCNs) to represent such relationships again**. We found in our experiments that using GCNs does not improve performance compared to using GA. This is because the core idea of GCNs is to infer the relationship between nodes based on the input adjacency matrix and node embeddings and then update the node embeddings based on this relationship. However, the function of inferring the relationship between nodes is precisely accomplished by BGI and EGT. The EmoG has adequately represented the relationship between nodes under specific emotions, and the GA operation is also more efficient than the convolution layers or graph neural networks. Therefore, we simplify the GCNs to GA that retain only the function of updating node embeddings.
>
>
> ## Answer to Question 2:
>
> If we understand the reviewer correctly, the reviewer is concerned that there is a step in BGI that couples $n$ relationships between multi-modal signals. This step constructs multiple graphs, thus incurring a substantial computational burden. However, this step does not construct multiple graphs and, therefore, does not increase the computational burden.
>
>
> **The BGI does not directly align the graphs between domains, but rather the edge existence probability distributions, so it doesn't construct multiple graphs**. In the paper, we mentioned:
>
> > From this, we define the prior HetG edge distribution from the source domain as follows:
> > $$P(\text{HetG}|E_s) \sim \text{BIN}(n, p_s)$$
>
> where $p_s \in \mathbb{R} ^{B \times N_{e} \times 1}$ is computed by the network and mathematically represents the probability of the existence of each of the $N_e$ edges. It is much smaller than the edge embedding matrix $E_s\in \mathbb{R}^{B \times N_{e} \times D_e}$.
>
> To align domains by such probability distributions, we design the BGI Loss (essentially Kullback-Leibler Divergence (KLD)), which is used to minimize the divergence between the two probability distributions. However, as the reviewer is concerned, the direct computation of this KLD imposes a substantial computational burden due to the presence of $n$. To make this computation possible in the network, we propose Theorem 2, which mathematically computes an upper bound for this KLD:
> > $$\mu_{lt} \log \frac{\mu_{lt}+\epsilon}{p_s+\epsilon} +(1-\mu_{lt}) \log \frac{1-\mu_{lt}+{\mu_{lt}}^2 / 2+\epsilon}{1-p_s+{p_s}^2 / 2+\epsilon}$$
>
> which is not directly related to $n$. The computational complexity of this equation is $O(B \cdot N_e)$, which will not impose a significant computational burden.
>
>
> ## Answer to Question 3:
>
> Yes, the class imbalance issue is a common and challenging problem in cross-subject EEG ER. Detailed investigation of the imbalance issue will be part of our future effort. Actually, our method achieves the second-best results regarding the F1 score among all methods in most cases. For a better illustration, we updated Table 1 and highlighted the top 3 performances to show the advantages of our method.
>
> ## Answer to Question 4:
>
> See the answer in Weakness 3.

---

> ### Author Response · Authors · 2023-11-19
> **Response to Reviewer QvEM (4/4)**
>
> ## Answer to Question 5
>
> Thank you for bringing this to our attention! We agree that the heterogeneity (e.g., EEG, ECG) mentioned in our paper can be regarded as multi-variant time series data.
>
> The heterogeneous data in our paper is to emphasize the diversity of physiological signals captured by different sensors. The term "heterogeneity" is commonly used in multi-modal or EEG ER [5][6][7]. The multi-variant time series data is the sequential observations (e.g., physiological signals and vital signs) that may be irregularly sampled by sensors [9]. Therefore, the heterogeneous data and multi-variant time series data **have little differences in data format**.
>
> As suggested by the reviewer, the baseline methods for learning multi-variant time series can also be used for the ER task. We adopt two graph structure based methods [8][9] as our new baselines. Their preliminary experimental results are as follows:
>
> | Method | DEAP |  |  |  | DREAMER |  |  |  |
> | :---: | :---: | :---: | :---: | :---: | :---: | :---: | :---: | :---: |
> |  | Arousal |Arousal  | Valence | Valence | Arousal | Arousal | Valence |Valence   |
> |  | Accuracy | F1 Score | Accuracy | F1 Score | Accuracy | F1 Score | Accuracy | F1 Score |
> | MTGNN [8] | $67.46 \pm 11.51$ | $63.03 \pm 12.19$ | $64.77 \pm 7.98$ | $67.24 \pm 8.33$ | $66.66 \pm 9.54$ | $66.24 \pm 11.5$ | $63.35 \pm 6.29$ | $64.01 \pm 9.39$ |
> | RAINDROP [9] | $66.06 \pm 10.11$ | $63.7 \pm 12.43$ | $65.59 \pm 7.38$ | $64.29 \pm 7.98$ | $65.74 \pm 8.99$ | $62.17 \pm 10.82$ | $65.85 \pm 7.61$ | $62.44 \pm 8.07$ |
> | Our VBH-GNN | **73.5** $\pm$ **7.22** | **71.53** $\pm$ **10.86** | **71.21** $\pm$ **6.41** | **71.85** $\pm$ **7.38** | **70.64** $\pm$ **7.74** | **69.66** $\pm$ **9.51** | **73.38** $\pm$ **4.21** | **69.08** $\pm$ **6.98** |
>
>
>
> We observe that **the baselines designed for multi-variant time series data perform not that good when applied to emotion recognition on our heterogeneous data. This is because the heterogeneous data and the multi-variant time series data have different characteristics, such as data sparsity and sampling rate.**
>
>
>
>
> [5] Ziyu Jia, et al. "HetEmotionNet: two-stream heterogeneous graph recurrent neural network for multi-modal ER." ACM MM, 2021.
>
> [6] Linlin Gong, et al. "Emotion recognition from multiple physiological signals using intra-and inter-modality attention fusion network." Digital Signal Processing, 2024.
>
> [7] Wei Li, et al. "Can emotion be transferred?—A review on transfer learning for EEG-Based Emotion Recognition." IEEE TCDS, 2021.
>
> [8] Zonghan Wu, et al. "Connecting the dots: Multivariate time series forecasting with graph neural networks." ACM SIGKDD, 2020.
>
> [9] Xiang Zhang, et al. "Graph-Guided Network for Irregularly Sampled Multivariate Time Series." ICLR, 2022.
>
> We hope that our answers clarify your concerns. Thank you for your time and valuable feedback! We are glad to have any further discussion.

---

> > ### Comment · Reviewer_QvEM · 2023-11-23
> > **Increase my rate to 6**
> >
> > I want to thank the authors for addressing my concerns. I'll increase my rate to above the threshold.

---

> > > ### Author Response · Authors · 2023-11-23
> > > **Response to Reviewer QvEM**
> > >
> > > We are glad that our responses addressed the reviewer's concerns. We greatly appreciate the reviewer for taking the time to provide thoughtful feedback and raise the score.

---

### Meta-Review · Area_Chair_4w4t · 2023-12-06

**Metareview:**

The authors introduce a new approach to physiological emotion recognition attempts to align source and target distributions by multi-modal spatial-temporal relationships .  The method they introduce is called "Variational Bayesian Heterogeneous Graph Neural Networks" (VBH-GNN).  It integrates the temporal and spatial modalities between domains and ER into one framework. Specifically, the paper contains two innovations: Bayesian Graph Inference (BGI) and Emotional Graph Transform (EGT). The BGI models multi-modal relationships as heterogeneous graphs (HetG) and aligns the relationships of domains via the edge distribution of HetG based on the Variational Bayesian theorem. EGT transforms the HetG into emotion-specific graphs (EmoG), further aligning the source and target domains while differentiating the relationships of modalities under different emotions. After the joint constraints of these two steps, the VBH-GNN can infer the domain-invariant multi-modal spatio-temporal relationships between source and target domains and
utilize this relationship to weight the signal feature and feed it to the classifier for cross-subject ER.

Strengths:

-The authors present extensive quantitative and qualitative experiments on multiple datasets.
-The results demonstrate that VBH-GNN significantly outperform previous state-of-the-art methods.
-The paper  "offers an in-depth analysis of the current challenges within the emerging domain of human emotion recognition using electroencephalogram (EEG), with a specific emphasis on cross-subject emotion recognition (ER)."
-The authors come up with an interesting method to combat heterogeneity across patients by using multiple modalities.
-The authors provide a rigorous proof and interpretability of their method.

Weaknesses:

Many weakness were addressed in the rebuttal:
- The experimental setting of dividing source and target domains is unrealistic.
-Statistical results in tables will be more convincing.
-In addition, the optimal results in Table 1 should all be highlighted.
-The quality Figure 2 needs to be improved.
-The explanation of the (Emotional graph transform) EGT step lacks depth. Although the authors demonstrate in their experiment (4.5) the difference between EGT and BGI, which is able to transform an intermodal heterogeneous graph into a more emotion-specific graph, the motivation for this step is not clear enough to me, and it seems more like a step based on experimental attempts to determine what to do; in other words, the authors seem to know what has to be done and how it should be done, but are unable to explain why it allows HetG to be transformed in an emotionally weighted way

**Justification For Why Not Higher Score:**

I do not believe that this paper will have wide impact, but it will be of interest to specific people so I feel a poster is sufficient.

**Justification For Why Not Lower Score:**

There is no reason not to accept this paper.  It is sound and will be important to researchers in a particular sub field.

---

### Decision · Program_Chairs · 2024-01-16

Accept (poster)